# *ERBB2* mRNA Expression and Response to Ado-Trastuzumab Emtansine (T-DM1) in HER2-Positive Breast Cancer

**DOI:** 10.3390/cancers12071902

**Published:** 2020-07-14

**Authors:** Gaia Griguolo, Fara Brasó-Maristany, Blanca González-Farré, Tomás Pascual, Núria Chic, Tamara Saurí, Ronald Kates, Oleg Gluz, Débora Martínez, Laia Paré, Vassilena Tsvetkova, David Pesantez, Maria Vidal, Barbara Adamo, Montserrat Muñoz, Patricia Galván, Laura Barberá, Miriam Cuatrecasas, Mathias Christgen, Hans Kreipe, Inés Monge-Escartín, Patricia Villagrasa, Dolors Soy, Tommaso Giarratano, Maria Vittoria Dieci, Pierfranco Conte, Nadia Harbeck, Valentina Guarneri, Aleix Prat

**Affiliations:** 1Division of Medical Oncology 2, Istituto Oncologico Veneto IOV—IRCCS, 35128 Padova, Italy; gaia.griguolo@iov.veneto.it (G.G.); tommaso.giarratano@iov.veneto.it (T.G.); mariavittoria.dieci@unipd.it (M.V.D.); pierfranco.conte@unipd.it (P.C.); valentina.guarneri@unipd.it (V.G.); 2Department of Surgery, Oncology and Gastroenterology, University of Padova, 35124 Padova, Italy; vassilena.tsvetkova@gmail.com; 3Department of Medical Oncology, Hospital Clínic de Barcelona, 08036 Barcelona, Spain; fbraso@clinic.cat (F.B.-M.); topascual@clinic.cat (T.P.); chic@clinic.cat (N.C.); sauri@clinic.cat (T.S.); demartinez@clinic.cat (D.M.); Pesantez@clinic.cat (D.P.); MJVIDAL@clinic.cat (M.V.); ADAMO@clinic.cat (B.A.); MMUNOZ@clinic.cat (M.M.); galvan@clinic.cat (P.G.); LBARBERA@clinic.cat (L.B.); 4Translational Genomics and Targeted Therapeutics in Solid Tumors, August Pi i Sunyer Biomedical Research Institute (IDIBAPS), 08036 Barcelona, Spain; MBGONZAL@clinic.cat; 5Department of Pathology, Hospital Clínic de Barcelona, 08036 Barcelona, Spain; MCUATREC@clinic.cat; 6SOLTI breast cancer cooperative group, 08008 Barcelona, Spain; Laia.pare@gruposolti.org (L.P.); patricia.villagrasa@gruposolti.org (P.V.); 7The West German Study Group, 41061 Mönchengladbach, Germany; ronald.kates@t-online.de (R.K.); oleg.gluz@wsg-online.com (O.G.); 8Ev. Hospital Bethesda, Breast Center Niederrhein, 41061 Mönchengladbach, Germany; 9University Clinics Cologne, 50937 Cologne, Germany; 10Medical School Hannover, Institute of Pathology, 30625 Hannover, Germany; christgen.matthias@mh-hannover.de (M.C.); Kreipe.Hans@mh-hannover.de (H.K.); 11Pharmacy Department, Division of Medicines, Hospital Clínic de Barcelona, 08036 Barcelona, Spain; monge@clinic.cat (I.M.-E.); DSOY@clinic.cat (D.S.); 12Breast Center, Department of Gynecology and Obstetrics, University of Munich (LMU) and CCCLMU, 80337 Munich, Germany; Nadia.Harbeck@med.uni-muenchen.de

**Keywords:** *ERBB2* mRNA, HER2-positive breast cancer, T-DM1, antibody-drug conjugates

## Abstract

Trastuzumab emtansine (T-DM1) is approved for the treatment of human epidermal growth factor receptor 2 (HER2)-positive (HER2+) metastatic breast cancer (BC) and for residual disease after neoadjuvant therapy; however, not all patients benefit. Here, we hypothesized that the heterogeneity in the response seen in patients is partly explained by the levels of human epidermal growth factor receptor 2 gene *(ERBB2)* mRNA. We analyzed *ERBB2* expression using a clinically applicable assay in formalin-fixed paraffin-embedded (FFPE) tumors (primary or metastatic) from a retrospective series of 77 patients with advanced HER2+ BC treated with T-DM1. The association of *ERBB2* levels and response was further validated in 161 baseline tumors from the West German Study (WGS) Group ADAPT phase II trial exploring neoadjuvant T-DM1 and 9 in vitro BC cell lines. Finally, *ERBB2* expression was explored in 392 BCs from an in-house dataset, 368 primary BCs from The Cancer Genome Atlas (TCGA) dataset and 10,071 tumors representing 33 cancer types from the PanCancer TCGA dataset. High *ERBB2* mRNA was found associated with better response and progression-free survival in the metastatic setting and higher rates of pathological complete response in the neoadjuvant setting. *ERBB2* expression also correlated with in vitro response to T-DM1. Finally, our assay identified 0.20–8.41% of tumors across 15 cancer types as *ERBB2*-high, including gastric and esophagus adenocarcinomas, urothelial carcinoma, cervical squamous carcinoma and pancreatic cancer. In particular, we identified high *ERBB2* mRNA in a patient with HER2+ advanced gastric cancer who achieved a long-lasting partial response to T-DM1. Our study demonstrates that the heterogeneity in response to T-DM1 is partly explained by *ERBB2* levels and provides a clinically applicable assay to be tested in future clinical trials of breast cancer and other cancer types.

## 1. Introduction

Trastuzumab emtansine (T-DM1) is an antibody–drug conjugate (ADC) linking the anti-HER2 (human epidermal growth factor receptor 2) monoclonal antibody trastuzumab to a microtubule inhibitor, DM1. T-DM1 is approved in several countries as single-agent treatment for HER2+ metastatic breast cancer (BC) patients previously treated with trastuzumab and a taxane. In the phase III randomized EMILIA trial, T-DM1 was compared with capecitabine plus lapatinib in previously treated (trastuzumab and taxane) HER2+ metastatic BC patients [1]. In the phase III randomized TH3RESA trial, enrolling patients previously treated with trastuzumab and lapatinib in the advanced setting, T-DM1 was compared with treatment of physician’s choice [2]. Treatment with T-DM1 was associated with a significant improvement in both progression-free survival (PFS) and overall survival (OS) in both trials. Recently, trastuzumab emtansine (T-DM1) has also been approved (Food and Drug Administration and European Medicines Agency) for the treatment of residual invasive disease after neoadjuvant treatment for HER2+ BC. In fact, the phase III randomized KATHERINE trial, which enrolled HER2+ BC who were found to have residual invasive disease (breast or axilla) after receiving neoadjuvant therapy containing at least a taxane and trastuzumab, demonstrated a clear improvement in invasive disease-free survival for patients who were switched to T-DM1 instead of continuing trastuzumab [3].

HER2-positivity is currently defined by semi-quantitative methods such as immunohistochemistry (IHC) and in situ hybridization (ISH). This definition was originally designed to predict benefit from trastuzumab in advanced and adjuvant trials [4,5], and has remained the definition for the development of the rest of anti-HER2 therapies, including T-DM1. However, the classic definition of HER2-positive breast cancer has been recently been challenged. For example, the presence of HER2 intratumor heterogeneity plays a significant role in modulating response to anti-HER2 treatments and is associated with worse patient outcomes, in terms of shorter disease free survival and overall survival [6,7,8], an aspect scarcely accounted for by semi-quantitative evaluations; at the same time, new HER2-targeted ADCs approved [9] or currently under investigation in phase III trials for HER2+ advanced BC have been reported to be active in HER2-low BC (defined as IHC 1+ or 2+/non amplified) [10,11,12], thus raising the question if different definitions of HER2-positivity should be adopted for each HER2-targeted agent. 

Here, we hypothesized that quantitative measurements of HER2, such as *ERBB2* mRNA expression, might further help better identify within HER2+ metastatic BC patients those who will benefit from T-DM1. 

## 2. Results

### 2.1. ERBB2 mRNA in Advanced HER2+ BC Treated with T-DM1 

Seventy-seven consecutive patients diagnosed with HER2+ advanced BC and treated with T-DM1 at Hospital Clínic of Barcelona (HCB) and Istituto Oncologico Veneto (IOV) in Padova were evaluated. Demographic and disease characteristics of these patients are presented in Table 1. Briefly, all patients were pre-treated with trastuzumab in the (neo)adjuvant or metastatic setting and most had received at least 1 line of HER2-targeted treatment for metastatic disease (median 1, range 0–4). In addition, twenty-eight patients (36%) presented brain metastases at time of initiation of T-DM1. Regarding T-DM1 efficacy, overall response rate (ORR) was 47% (6 complete and 30 partial responses) and median progression-free survival (PFS) was 5.3 months (95% CI 4.4–10.7). 

*ERBB2* mRNA expression was assessed in 77 tumor samples from patients with advanced HER2+ BC treated with T-DM1. After determination of an *ERBB2* mRNA cutoff predictive of response to T-DM1, several cohorts were evaluated. On one hand, the *ERBB2* mRNA assay was validated in 161 patients recruited in the WSG-ADAPT phase II neoadjuvant trial of T-DM1. On the other, *ERBB2* mRNA expression and response was evaluated in 9 in vitro BC cell lines. Finally, *ERBB2* mRNA expression was explored in 392 BCs from an in-house dataset, 368 primary BCs from the TCGA BC dataset, and 10,071 tumors representing 33 cancer types from the PanCancer TCGA dataset (Figure 1).

A large range of *ERBB2* mRNA expression was observed (log2 median 2.98; interquantile range 1.60–3.91). As expected, the expression of *ERBB2* varied according to HER2 IHC expression (0, 1+, 2+ and 3+) (Appendix A). Compared to 0–2+, the expression in 3+ tumor samples was increased by 5.7-fold. Of note, although all patients were classified as HER2+ tumors by guidelines [4], based on clinical history and previous assessment of HER2 status on tumor samples, and were eligible for treatment with T-DM1, 8 of the 74 samples that were re-analyzed for HER2 had an IHC HER2 result of 0 (*n* = 5) or 1+ (*n* = 3). Four of these 8 samples were tested for HER2 amplification by ISH, and HER2 was found amplified in 3 cases and non-amplified in 1 sample. 

The clinicopathological variables associated with response (i.e., partial and complete) to T-DM1 were: negative hormone-receptor status, lower number of prior lines of HER2-targeted therapy in the metastatic setting, higher HER2 IHC expression (i.e., 3+ vs. 0–2+) and higher *ERBB2* mRNA as a continuous variable (Table 2). The overall response rate in HER2 3+ was 62.0% compared to 20.8% in the 0–2+ group (odds ratio = 1.84, 95% confidence interval [CI] 1.26–2.69, *p* = 0.002). However, only *ERBB2* mRNA expression (as a continuous variable) and number of prior HER2-targeted lines, but not HER2 IHC expression or hormone-receptor status, were found independently associated with response (Table 2). 

### 2.2. Identification of an Optimized ERBB2 mRNA Cutoff

Using T-DM1 response data in advanced HER2+ BC (i.e., PR and CR vs. stable disease [SD] and progression of disease [PD]), an optimized *ERBB2* mRNA cutoff was identified based on Fisher’s exact test. This cutoff maximized the area under the curve (AUC) of the receiver operating characteristic (ROC) curve (AUC = 0.701, sensitivity: 100%, specificity: 41.5%) (Figure 2a). The cutoff was *ERBB2* log2 value = 1.483 and it classified 60 tumors (77.9%) as *ERBB2*-high and 17 (22.1%) as *ERBB2*-low (Figure 2b). As expected, the *ERBB2*-high group had a response rate of 60%, whereas the response rate in the *ERBB2*-low group was 0%. ERBB2-high, as compared to ERBB2-low, was significantly associated with higher response rates, both in the HER2 IHC 3+ subgroup (67% vs. 0%, Fisher exact *p* = 0.017) and in the HER2 IHC 0–2+ subgroup (42% vs. 0%, Fisher exact *p* = 0.037).

Finally, the *ERBB2*-high group had a better PFS compared to the *ERBB2*-low group (median PFS 6.2 months vs. 2.93 months; hazard ratio = 0.36, 95% CI 0.20–0.65, *p* = 0.001) (Figure 2c), even after correction by number of prior lines of HER2-targeted therapy in the metastatic setting (hazard ratio = 0.38, 95% CI 0.21–0.70, *p* = 0.002).

### 2.3. Validation of ERBB2 mRNA Expression in Early-Stage HER2+ BC Treated with Neoadjuvant T-DM1

To further validate the *ERBB2*-based assay as a predictor of response to T-DM1 in the neoadjuvant setting, we assessed gene expression and pCR data from 158 patients treated with T-DM1 (alone or in combination with endocrine therapy) in the WSG-ADAPT HER2+/hormone receptor-positive (HR+) Phase II Trial [13]. Since gene expression in the WSG-ADAPT trial was determined using a different nCounter CodeSet and house-keeping gene list, we estimated where our *ERBB2* cutoff would fall. To accomplish this, we first determined the percentile of *ERBB2* mRNA expression corresponding to our *ERBB2* cutoff (i.e., 1.483) in 77 HER2+/HR+ primary tumors from our previously published PAMELA trial [14], since this cohort is similar to the WSG-ADAPT cohort. Our *ERBB2* cutoff corresponded to the 50th percentile in primary HER2+/HR+ tumors. 

We then applied the 50th percentile of *ERBB2* mRNA expression as the cutpoint to define *ERBB2*-high from *ERBB2*-low in the 161 patients of WSG-ADAPT trial. The overall pCR rate was 34.8% (56/161). When the *ERBB2* cutoff was evaluated, the pCR rate in the *ERBB2*-high group was 42.9% (33/77) and in the *ERBB2*-low group was 27.4% (23/84) (sensitivity = 58.9%; specificity = 58.1%; odds ratio = 2.0; *p*-value = 0.041). Altogether, these results confirmed a significant association between *ERBB2* mRNA levels and response to neoadjuvant T-DM1.

### 2.4. Exploring ERBB2 mRNA Expression and In Vitro Response to T-DM1 

Next, we evaluated the expression of *ERBB2* and the effects of T-DM1 across 6 HER2+ (HCC1954, ZR-75-30, BT-474, SK-BR3, HCC1569 and MDA-MB-453) and 3 HER2-negative BC cell lines (MCF7, T-47D and MDA-MB-468) (Figure 3a,b). *ERBB2* mRNA expression varied substantially across cell lines and similarly to patient’s tumors (interquartile range of 6.16 across all cell lines and 0.60 within the HER2+ cell lines). As expected, HER2+ cell lines showed higher *ERBB2* expression than the HER2-negative cell lines (mean 1.39 vs. −4.64, *p* = 0.0002) (Appendix A). HER2+ cell lines were also more responsive to T-DM1 than HER2-negative cell lines (Figure 3c and Appendix A), consistently with findings reported by others [15,16]. Response was defined as the decrease in cell viability (%) at 72 h of T-DM1 treatment. HER2+ cell lines showed greater response to T-DM1 than HER2-negative cell lines (mean response 54.71% vs. 5.24%, *p* = 0.008) (Appendix A). Importantly, we observed response to T-DM1 in all cell lines with *ERBB2* mRNA levels above the cutoff. In addition, correlation between *ERBB2* mRNA and T-DM1 response was observed across the 9 cell lines (coefficient = 0.7, *p* = 0.05) (Figure 3d). This correlation coefficient suggests that 76.6% of the differences in response across cell lines may be explained by *ERBB2* levels.

### 2.5. ERBB2 mRNA Expression in BC across the HER2 IHC-Based Groups

To determine the proportion of *ERBB2*-high tumors across the HER2 IHC groups, we analyzed a retrospective dataset of 392 tumor samples from HCB with both HER2 IHC status and gene expression data. As HER2 protein increased, *ERBB2* mRNA levels increased as well, and all possible comparisons (except for 1+ vs. 2+/non-amplified) of *ERBB2* expression between groups were statistically significant. According to our pre-established cutoff, the proportion of *ERBB2*-high across 0, 1+, 2+/ISH-negative, 2+/ISH-positive and 3+ was 0%, 1.1%, 0%, 9.38% and 76.17%, respectively (Figure 4a).

To provide more evidence of the association of *ERBB2* and HER2 IHC expression in BC, we explored 368 BCs from the TCGA dataset including *ERBB2* mRNA expression and HER2 IHC. Since the methodology to assess *ERBB2* mRNA expression in our dataset (nCounter) was different to the TCGA dataset (RNAseq), the range of *ERBB2* mRNA expression was different for each cohort. Therefore, we estimated where the pre-established cutoff would fall in the TCGA cohort. To do so, we calculated the median *ERBB2* mRNA log2 values for each IHC group in HCB and TCGA cohorts (Appendix A) and the Pearson correlation between the two cohorts (Appendix A). The proportion of *ERBB2*-high across 0, 1+, 2+/ISH-negative, 2+/ISH-positive and 3+ was 0%, 0.59%, 0%, 25% and 74.04%, respectively (Figure 4b). The correlation coefficient between these *ERBB2*-high proportions and the proportions found in our in-house dataset was 0.975.

### 2.6. ERBB2 mRNA Expression across Cancer Types

In order to determine the proportion of *ERBB2*-high tumors in other cancer types, we explored *ERBB2* expression for 10,071 tumors of different origins. According to our pre-established cutoff, *ERBB2*-high tumors were identified in 15 cancer types including: prostate cancer (0.2%), lung adenocarcinoma (0.59%), lung squamous cell carcinoma (0.83%), head and neck squamous cell carcinoma (0.97%), ovarian serous cystadenocarcinoma (1%), colon adenocarcinoma (1.14%), uterine carcinosarcoma (1.75%), uterine corpus endometrial carcinoma (2.09%), pancreatic adenocarcinoma (3.39%), rectum adenocarcinoma (3.9%), bladder urothelial carcinoma (3.93%), cervical squamous cell carcinoma (4.08%), esophageal adenocarcinoma (6.08%), stomach adenocarcinoma (6.32%) and breast cancer (9.41%) (Figure 5 and Appendix A). As expected, we identified a lower proportion of *ERBB2*-high tumors in each cancer type as compared to standard IHC/ISH definition of HER2-positivity, thus potentially selecting tumors sensitive to T-DM1 treatment even in cancer types generally not considered amenable to treatment with this agent.

### 2.7. ERBB2 mRNA Expression in HER2+ Gastric Cancer Treated with T-DM1

We retrospectively studied the case of a 42-year-old male with a gastroesophageal adenocarcinoma diagnosed at HCB. In September 2011, a total esophagectomy and cervical esophago-gastric anastomosis was performed. Ascites and pleural effusion positive cytology adenocarcinoma were observed after surgery, and the patient was diagnosed with a HER2 3+ pT2pN3pM1 stage IV gastroesophageal junction adenocarcinoma. The patient subsequently received 6 cycles of first-line treatment with cisplatin plus 5-fluorouracil and trastuzumab achieving a radiological CR.

After 10 months, the patient presented with lung and bone PD and was enrolled in the GATSBY phase II/III trial [17] and treated with T-DM1 (2.4 mg/kg weekly) monotherapy. Since July 2013, he received 8 cycles of T-DM1 and obtained a PR (Table 3 and Appendix A) and a time-to-progression of 5.4 months. Of note, median PFS and ORR in GATSBY’s T-DM1 arm was 2.7 months and 20.6%, respectively [17]. Concordant with the efficacy results obtained in our patient, *ERBB2* mRNA levels measured in the primary tumor were high (i.e., 2.99). Upon PD, the patient received third-line docetaxel monotherapy achieving SD until July 2014 when he presented with a central nervous system PD. The patient was lost to follow-up in November 2014.

## 3. Discussion

As an increasing number of HER2-targeted agents are becoming available in clinical practice, biomarkers are increasingly needed that can predict the response to specific anti-HER2 agents beyond the classic IHC/ISH definition of HER2-positivity. In this context, HER2 3+ tumors have been reported to benefit more from T-DM1 than other IHC groups in retrospective and prospective studies [7,18]. Moreover, benefit associated with the use of post-neoadjuvant T-DM1 as compared to trastuzumab in the KATHERINE trial appeared to more marked in HER2 3+ than in HER2 2+ tumors [19]. Greater benefit to T-DM1 in HER2 3+ tumors has also been reported in the KATE2 trial [20] and other cancer types [21,22].

*ERBB2* mRNA expression has been previously associated with a more pronounced T-DM1 benefit in several randomized clinical trials which tested the use of T-DM1 as compared to other HER2-targeted treatments in HER2+ metastatic BC [23,24,25]. In the randomized EMILIA trial, which compared T-DM1 and capecitabine-lapatinib (CL) for pretreated metastatic HER2+ BC patients, patients with tumor *ERBB2* mRNA levels above median showed a greater benefit from the use of T-DM1 in terms of ORR and overall survival (OS). However, T-DM1 treatment, compared with CL, reduced the risk of PD to a similar degree regardless of tumor *ERBB2* mRNA levels and tests for interaction between treatment and *ERBB2* mRNA levels were not statistically significant (*p* = 0.07). However, tests were exploratory and not powered to detect an interaction [23].

Moreover, data from the randomized TH3RESA trial, which compared T-DM1 vs. treatment of physician choice for pretreated metastatic HER2+ BC patients, confirmed that patients with higher *ERBB2* mRNA levels benefited more from T-DM1 than patients with lower *ERBB2* mRNA levels [24]. Similar results confirming an association between higher *ERBB2* mRNA levels and increased T-DM1 benefit, both in terms of ORR and PFS, were also reported for several randomized phase II trials [25,26,27]. Furthermore, the impact of *ERBB2* mRNA levels on T-DM1 benefit has been reported in other cancers beyond BC. A translational study [21] evaluating gastric cancer samples of the GATSBY trial [17] demonstrated more benefit to T-DM1 in terms of PFS in patients with tumors with higher *ERBB2* mRNA levels [21].

In addition, recently presented biomarker data from the post-neoadjuvant KATHERINE trial have been reported showing that patients with high *ERBB2* mRNA levels (above median) at surgery have a worse outcome than patients with low *ERBB2* mRNA levels when treated with adjuvant trastuzumab, but not when treated with adjuvant T-DM1. In fact, while both patients with *ERBB2* mRNA levels above and below median levels benefited from switching to T-DM1, those with higher *ERBB2* mRNA levels (above median) benefited more from T-DM1 than those with lower *ERBB2* mRNA levels, potentially questioning the use of median mRNA expression value as cutoff in this setting [28].

Our results not only confirm the association between *ERBB2* mRNA expression and T-DM1 benefit in a more heterogeneous real-world setting, but also highlight the relevance of a quantitative method as a better method to predict response to T-DM1 by proposing a cutoff for selecting patients responsive to T-DM1 both in the metastatic and neoadjuvant settings. Other HER2-targeted ADCs are entering clinical practice, for instance DS-8201, which has shown activity in HER2-low advanced BC [12,29]. Therefore, we might expect that different *ERBB2* cutoffs will be needed for different ADCs. In this context, the use of a quantitative method such as *ERBB2* mRNA expression, which offers the opportunity to identify different cutoffs, might potentially improve treatment personalization. Moreover, a quantitative method as *ERBB2* mRNA expression might recapitulate tumor heterogeneity in a single, easily manageable assay. 

Our study has several limitations. First, the study cohort is retrospective and only involved a limited number of patients who were treated according to everyday clinical practice, thus being heterogeneous, both in previous lines of treatment received and in the kind of histological samples available (primary tumor vs. metastasis). Despite this, our analysis was able to clearly identify *ERBB2* mRNA expression as the main predictor of responsiveness together with number of previous lines of HER2-targeted treatment. Moreover, we validated *ERBB2* mRNA as a predictor of response in the neoadjuvant setting. This highlights the clinical importance of *ERBB2* mRNA expression. Second, we could not address if the biomarker works better when primary or metastatic tumor samples are used. Third, our data in non-BC-types is currently in the hypothesis-generating stage. 

## 4. Materials and Methods 

### 4.1. Patient Datasets and Tumor Samples

This study analyzed a retrospective cohort of 77 HER2+ (as defined by standard guidelines [4]) advanced/metastatic BC patients treated with T-DM1 between January 2013 and November 2019 in two independent institutions: Hospital Clínic of Barcelona (HCB) (*n* = 26) and Istituto Oncologico Veneto (IOV) in Padova (*n* = 51). One formalin-fixed paraffin-embedded (FFPE) tumor sample per patient was selected: if available, a biopsy of metastatic site nearest in time to start of T-DM1 was preferred (*n* = 38); otherwise primary tumor sample was used (*n* = 39), favouring pre-treatment biopsy over surgical sample for patients treated with neoadjuvant therapy. Gene expression was also assessed in 77 primary HER2+/HR+ BC of the PAMELA trial [14], and gene expression and pathological complete response (pCR) data was evaluated in 161 HER2+ primary samples of the T-DM1 arms (A&B) of the WSG-ADAPT HER2+/HR+ Phase II Trial [13]. In addition, we evaluated 392 primary BCs from HCB with available HER2 IHC status and gene expression data evaluated at the nCounter platform; 368 primary BCs from The Cancer Genome Atlas (TCGA) with HER2 IHC status and *ERBB2* RNASeqv2 data; and 10,071 TCGA pan-cancers with *ERBB2* RNASeqv2 data. Finally, we analyzed a primary tumor sample of a patient with advanced gastric cancer treated with T-DM1 in the GATSBY trial [17] at HCB.

### 4.2. In Vitro Cell Lines and T-DM1

The BC cell lines BT-474, HCC1569, HCC1954, MCF7, MDA-MB-453, MDA-MB-468, SK-BR3, T-47D and ZR-75-30 were purchased from the American Type Culture Collection. All cell lines were maintained as recommended by the suppliers. T-DM1 was provided as remnant of the product used in common clinical practice by the oncology pharmacy Service at HCB.

### 4.3. HER2 Immunohistochemistry and Fluorescent In Situ Hybridization

HER2 status was re-assessed in 74 of 77 FFPE tumors of the T-DM1 HCB/IOV cohort and FFPE BC cell line pellets by either IHC and/or in situ hybridization (ISH) according to the American Society of Clinical Oncologists (ASCO)/College of American Pathologists (CAP) guidelines. IHC was performed on 2-μm-thick sections using anti-HER-2/neu (4B5) Rabbit Monoclonal Primary Antibody kit (Ventana Medical Systems Inc., Oro Valley, AZ, USA) and ISH for HER2 was performed on 4-μm-thick sections using the FDA-approved XL ERBB2 (HER2/NEU) AMP (MetaSystems Probes, Altlußheim, Germany) according to manufacturer’s instructions. 

### 4.4. In Vitro Cell Viability Assay

BC cell lines were plated in triplicate at 4000 cells/well in 96-well plates. Cells were then treated with 1.25 μg/mL T-DM1. Cell viability was determined 72 h after treatment using CellTiter 96 AQueous One Solution Cell Proliferation Assay (MTS) (Promega Corporation, Madison, Wisconsin, USA) following the manufacturer’s instructions, and quantified using the Gen5 Microplate Reader and Imager Software (BioTek, Winooski, VT, USA). Data were analyzed using GraphPad Prism 5 software (GraphPad, San Diego, CA, USA).

### 4.5. RNA Extraction

RNA samples were extracted from biopsy and surgical tumor FFPE material using the High Pure FFPET RNA isolation kit (Roche) following manufacturer’s protocol. FFPE slides with at least 10% tumor cells and 4 mm^2^ of tissue were used for each tumor specimen, and macrodissection was performed to avoid contamination with normal breast tissue if needed. Cell line RNA samples were extracted using the RNeasy Mini Kit (Qiagen, Hilden, Alemanya). RNA samples were quantified at the NanoDrop spectrophotometer (Thermo Fisher Scientific, Waltham, MA, USA).

### 4.6. Gene Expression Analysis

The nCounter platform (NanoString Technologies, Seattle, WA, USA) was used to analyze RNA samples from tumors and cell lines. A minimum of 100 ng of total RNA was used to measure the expression of 50 genes of the PAM50 intrinsic subtype predictor assay and 5 housekeeping genes (*ACTB, MRPL19, PSMC4, RPLP0,* and *SF3A1*). Expression counts were then normalized using the nSolver 4.0 software (nanoString, Seattle, WA, USA) and custom scripts in R 3.4.3 (R Foundation, Vienna, Austria) [30].

### 4.7. Statistical Analysis

Univariate and multivariable logistic regression analyses were used to investigate the association of each variable with overall response. Odds ratios and 95% confidence intervals (CIs) were calculated for each variable. An optimized cutoff of gene expression was identified to predict overall response. Univariate and multivariable Cox proportional hazard regression analyses were performed to investigate the association of each variable with PFS. All statistical analyses were carried out using the R software version 3.4.3.

## 5. Conclusions

To conclude, our study presents a clinically applicable assay to help identify patients most likely to benefit from T-DM1, regardless of HER2 status. In addition, the assay could help identify patients most likely to benefit from other HER2-targeted ADCs across cancer types.

## Figures and Tables

**Figure 1 cancers-12-01902-f001:**
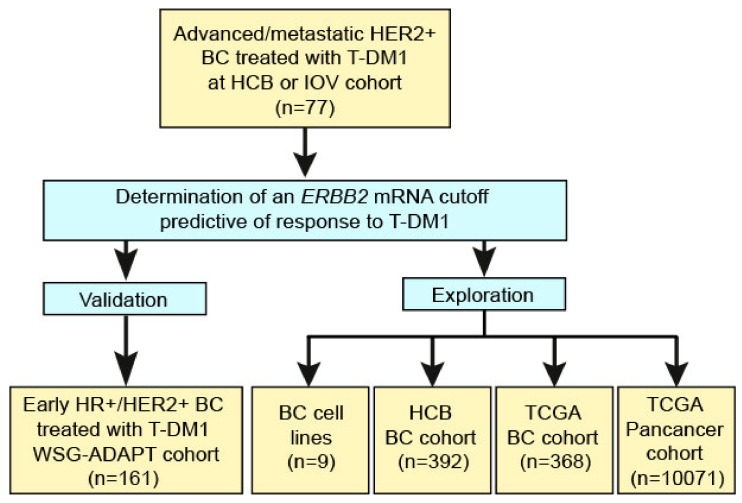
Workflow of the study.

**Figure 2 cancers-12-01902-f002:**
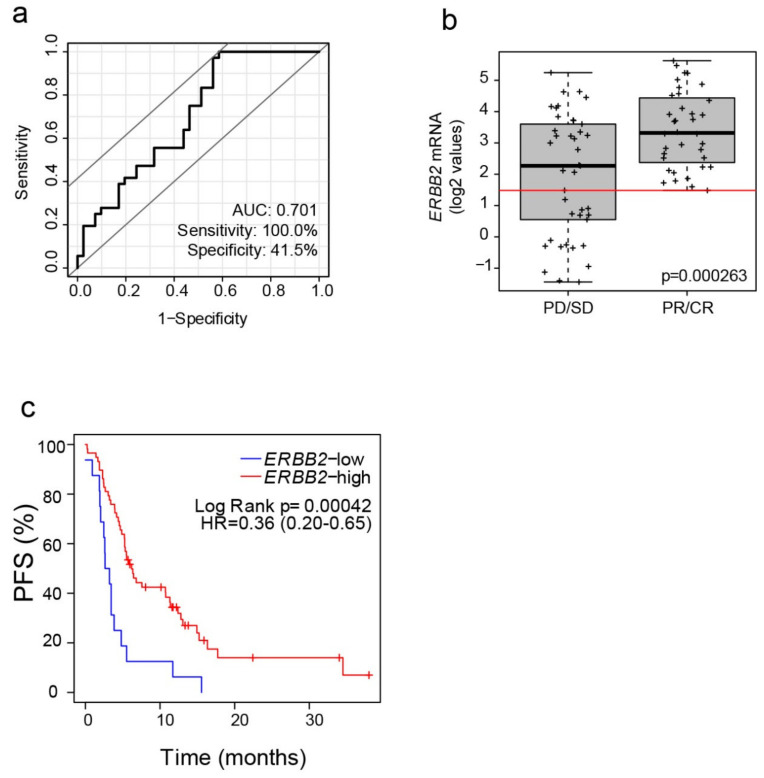
*ERBB2* mRNA expression predicts T-DM1 response and survival in metastatic HER2+ BC. (**a**) ROC curve to identify *ERBB2* mRNA cutoff of response to T-DM1. (**b**) *ERBB2* mRNA levels in patients with progressive disease (PD) or stable disease (SD) (*n* = 41) vs. patients achieving complete response (CR) or partial response (PR) (*n* = 36). *p*-value was determined using a two-tailed unpaired *t*-test. The *ERBB2* mRNA cutoff is shown as a red line. (**c**) Kaplan–Meier estimate of progression-free survival (PFS) using the *ERBB2* mRNA cutoff.

**Figure 3 cancers-12-01902-f003:**
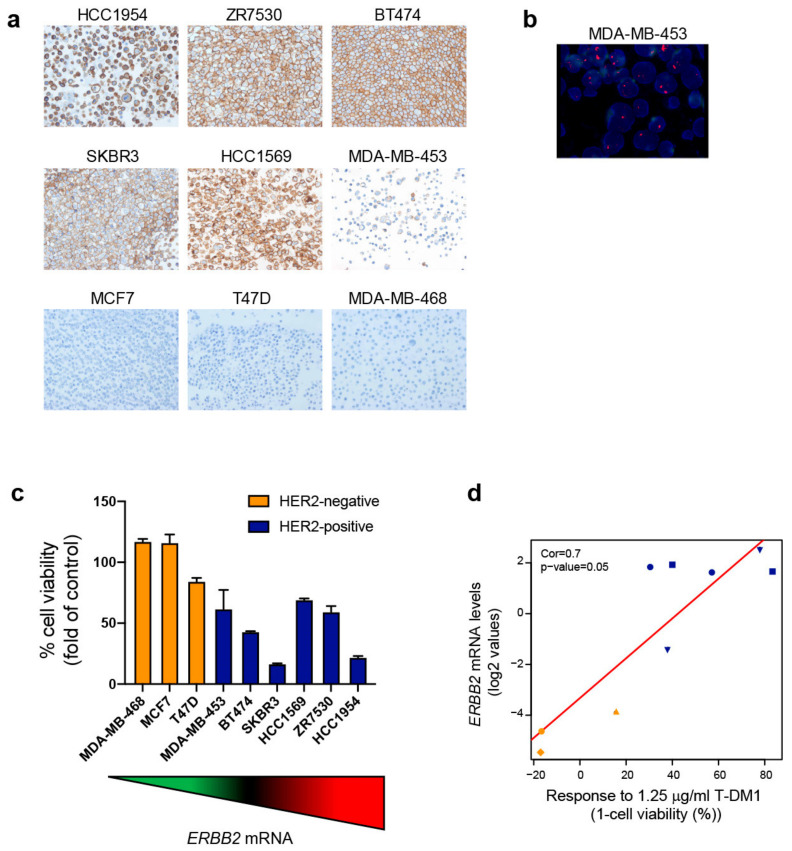
*ERBB2* mRNA expression correlates with response to T-DM1 in cell lines. (**a**) Images of HER2 expression by IHC in 9 BC cell lines (40×). (**b**) Image of HER2 amplification by ISH (100×). (**c**) Cell viability of 9 BC cell lines upon 72 h of treatment with 1.25 μg/mL T-DM1. Data points represent the mean; error bars represent the standard error of the mean of 3 independent experiments. (**d**) Spearman correlation between *ERBB2* mRNA expression and response to 1.25 μg/mL T-DM1 expressed as 1-cell viability (%).

**Figure 4 cancers-12-01902-f004:**
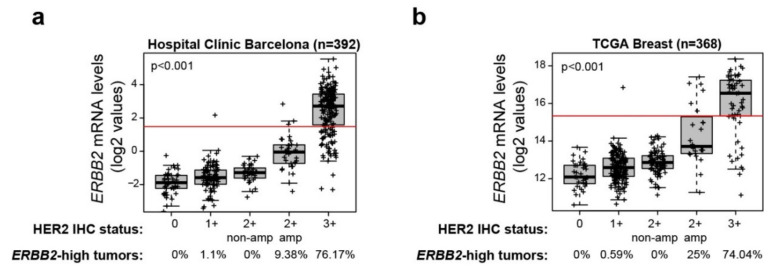
*ERBB2* mRNA expression in BC across the HER2 IHC groups. Distribution of *ERBB2* mRNA levels across HER2 IHC subgroups of the primary BC (**a**) HCB and (**b**) TCGA datasets. The proportion of *ERBB2*-high tumors is indicated as defined by the *ERBB2* mRNA cutoff (shown as a red line). *p*-values were determined using one-way analysis of variance.

**Figure 5 cancers-12-01902-f005:**
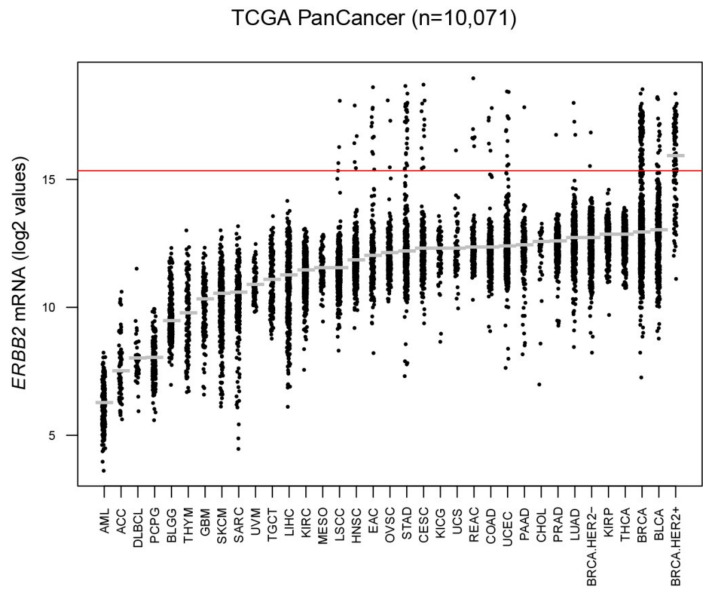
*ERBB2* mRNA expression across cancer types. Distribution of *ERBB2* mRNA levels across cancer-types TCGA datasets. The *ERBB2* mRNA cutoff is shown as a red line. Abbreviations are shown in Appendix A.

**Table 1 cancers-12-01902-t001:** Patient characteristics.

Characteristics	*n* = 77
Median age at BC diagnosis, years (range)	49 (27–88)
Median age at start of T-DM1, years (range)	51 (35–93)
Histology: Ductal	71 (92%)
Lobular/other	5 (6%)
NA	1 (1%)
Histologic Grade: G1	2 (3%)
G2	10 (13%)
G3	28 (36%)
NA	37 (48%)
Hormone-receptor: positive	46 (60%)
negative	31 (40%)
HER2 IHC status: IHC 0	5 (6%)
IHC 1+	3 (4%)
HC 2+	16 (21%)
IHC 3+	50 (65%)
NA *	3 (4%)
HER2 ISH status in HER2 IHC 2+ cases: Amplified	15 (94%)
Not evaluable *	1 (6%)
HER2 ISH status in HER2 IHC 0/1+ cases: Amplified	3 (38%)
Non-amplified *	1 (12%)
Not available *	4 (50%)
Previous (neo)adjuvant treatment	44 (57%)
Median number previous lines HER2-targeted	1 (0–4)
treatment for metastatic disease (range)	
Previously received:	
Pertuzumab-trastuzumab	31 (40%)
Trastuzumab	41 (53%)
Lapatinib	14 (24%)
Visceral metastases at start of T-DM1	66 (86%)
Brain metastases at start of T-DM1	28 (36%)
Concomitant endocrine treatment during T-DM1	17 (22%)

* These cases were confirmed to be HER2-positive in other tumor samples and treated with T-DM1 according to clinical practice.

**Table 2 cancers-12-01902-t002:** Univariable and multivariable logistic regression analyses of overall response.

Clinicopathological Variable		Univariate		Multivariable	
		Odds Ratio (95%CI)	*p*	Odds Ratio (95%CI)	*p*
Hormone-receptor status	negative	ref	0.038	ref	0.152
	positive	0.37 (0.14–0.95)		0.39 (0.11–1.41)	
De-novo metastatic disease	no	ref	0.990		
	yes	0.99 (0.40–2.49)			
Visceral disease	no	ref	0.577		
	yes	0.69 (0.19–2.50)			
Brain involvement	no	ref	0.966		
	yes	0.98 (0.39–2.49)			
HER2 IHC	≤2+	ref	0.002	ref	0.257
	3+	1.84 (1.26–2.69)		1.32 (0.82–2.13)	
*ERBB2* (continuous)		1.73 (1.25–2.39)	0.001	1.95 (1.22–3.12)	0.006
Prior lines HER2-targeted therapy	0–1	ref	0.009	ref	0.002
	≥2	0.06 (0.01-0.50)		0.02 (0.002-0.23)	

**Table 3 cancers-12-01902-t003:** Response evaluation criteria in solid tumors (RECIST) table for a gastric cancer case before and during T-DM1 treatment.

Target Lesion	Screening	Pre Cycle 3	Pre Cycle 5	Pre Cycle 7	Pre Cycle 9
Right upper lobe lung metastasis	14 mm	8 mm	8 mm	8 mm	10 mm
Left upper lobe lung metastasis	10 mm	10 mm	10 mm	10 mm	12 mm
Mesentheric adenopathy	17 mm	11 mm	11 mm	11 mm	18 mm
Retroperitoneal adenopathy	18 mm	9 mm	9 mm	9 mm	17 mm
Total	59 mm	38 mm	38 mm	38 mm	57 mm
Response	NA	39% reduction	39% reduction	39% reduction	36% increase
	NA	PR	Maintained PR	Maintained PR	PD

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
