# Peer review of "ERBB2* mRNA Expression and Response to Ado-Trastuzumab Emtansine (T-DM1) in HER2-Positive Breast Cancer"

_cancers, 2020, doi:10.3390/cancers12071902_

Round 1

Reviewer 1 Report

The manuscript “ERBB2 mRNA expression and response to ado-trastuzumab emtansine (T-DM1) in HER2-positive breast cancer” is aimed at the investigation of the association of ERBB2 levels and response to T-DM1 in HER2+ metastatic BC.

The authors present interesting data showing that the heterogeneity in response to T-DM1 may be partly explained by ERBB2 levels. Even though, the study has several limitations, as correctly reported in the discussion section of the manuscript, this study provides a quantitative method to predict response to T-DM1 both in the metastatic and neoadjuvant settings. The work seems well executed and the conclusions are of general interest and the manuscript should be publishable subsequent to some consideration of the following minor comments.

Minor concerns

One clinicopathological variables associated with response to T-DM was a negative hormone-receptor status.

The ERBB2 mRNA value as a predictor of response to T-DM1 was validated in 161 tumors from the HER2+/Hr+ of WSG-ADAPTphaseII trial. What about the HER2+/HR-cohorts?

In addition, the authors show that HER2+ cell lines give a greater response to T-DM1 than HER2-negative cell lines. Among the HER2+ cell lines, the SKBR3 and HCC1954 cells were more responsive to T-DM1 than HCC1569 or ZR7539 cells. Is there a correlation between ERBB2 mRNA and T-DM1 response in these HER2+ cell lines? The greater ERBB2 mRNA the major T-DM1 response?

Which is ERBB2 mRNA expression in these cell lines after T-DM1 treatment?

Reviewer 2 Report

In the present manuscript, the authors propose that quantitative measurements of ERBB2 mRNA expression can help in identifying HER2+ cancer patients who will benefit from T-DM1. The authors analyzed ERBB2 mRNA levels from 77 patients with advanced HER2+ breast cancer treated with T-DM1, and from 161 baseline tumors from the WGS-ADAPT phase II trial exploring neoadjuvant T-DM1. The authors showed that high ERBB2 mRNA was associated with better response and progression-free survival in the metastatic setting and higher rates of pathological complete response was in the neoadjuvant setting.

Further novelty of the manuscript is that the authors analyzed ERBB2 mRNA levels from 10,071 tumors representing 33 cancer-types from the PanCancer TCGA dataset (including 368 primary breast cancer), and showed important data.

The present manuscript is timely, and might have clinical relevance. In addition, the manuscript is well written, scientifically accurate and well-structured. I suggest the acceptance of this manuscript after its revision.

Comments:

Table S1.: the number of ERBB2-high in the Breast Invasive Carcinoma HER2 Positive group cannot be 6 – it is probably 66.

Table S1.: Can the authors show the ERBB2 mRNA expression data in the HER2 positive and HER2 negative subgroups of the Stomach Adenocarcinoma group?

Although T-DM1 showed growth inhibitory effect on preclinical models of HER2+ gastric cancer (Barok, Cancer Letters, 2011), the Phase 2/3 clinical trial in gastric cancer failed due to lack of efficacy of T-DM1 compared to chemotherapy (Thuss-Patience, Lancet Oncol, 2017). Still, HER2+ gastric cancer is a target indication for other ant-HER2 antibody-drug conjugates. Do the authors have additional ERBB2 mRNA expression data with HER2+ gastric cancer (more than the one case the authors showed)? Or with HER2+ gastric cancer cell lines?

Typo error:

Line 74 “development of the rest of anti-HER2 therapies, including T-DM1. However, the the classic…”

Line 97 HER2+ BC trated with T-DM1. After determination of an ERBB2 mRNA cutoff predictive of

Reviewer 3 Report

This manuscript describes that ERBB2 mRNA expression and response to T-DM1 in HER2-positive breast cancer in clinical and preclinical setting. The authors have some interesting results about the observed biomarker for T-DM1 sensitivity. However, there are a number of questions about your results.

1) In preclinical model in your study, ERBB2 mRNA expression and T-DM1 response is confusing (For example, high mRNA expression is not associated with response to T-DM1 in ZR7530 and HCC1569 cells)

2) In gastric cancer case, if the author had some information about ERBB2 expression after PD setting, it would be helpful to your study (For example, in PD setting, ERBB2 expression decrease compared to pretreatment setting)

Reviewer 4 Report

The assessment of HER2 expression by RNA, rather than IHC/FISH, is an important development and is of interest to both clinical and preclinical researchers.  Availability of RNA is more widespread and easier to obtain than IHC and FISH, allowing more extensive diagnostic, predictive and prognostic analyses of tumor samples.  Thus, this is an important area of progress for HER2-related research.

Specific parts of the manuscript that need to be addressed are annotated in the attached pdf.
